# The Impact of Air Pollution Perception on Urban Settlement Intentions of Young Talent in China

**DOI:** 10.3390/ijerph19031080

**Published:** 2022-01-19

**Authors:** Lianying Yao, Xuewen Li, Rongrong Zheng, Yiye Zhang

**Affiliations:** 1School of Economics, Zhejiang University of Technology, Hangzhou 310014, China; yaolianying@zju.edu.cn; 2Global Institute for Zhejiang Merchants Development, Zhejiang University of Technology, Hangzhou 310014, China; 3School of Public Affairs, Zhejiang University, Hangzhou 310058, China; 4School of Public Administration, Zhejiang University of Finance and Economics, Hangzhou 310018, China; 15068553536@163.com (R.Z.); zhangyiye@zufe.edu.cn (Y.Z.)

**Keywords:** air pollution perception, young talent, urban settlement intentions, residential satisfaction, place attachment

## Abstract

In recent years, with the public paying more and more attention to the problem of air pollution, the impact of air quality on migration has gradually become a growing concern. However, in the current context of cities’ efforts to “attract talent” in China, research on the impact of air pollution on the flow or dwelling willingness of young talent is relatively limited. Based on the theory of planned behavior and from the perspective of subjective perception, this paper uses a regulated model to explore the impact mechanism of air pollution perception on young talent urban settlement intentions. Taking Hangzhou as a case, this study surveyed 987 individuals who were classified as young talent to explore the impact of air pollution perception on urban settlement intentions in China. The research shows that air pollution perception has a significant impact on young talent urban settlement intentions, and this impact is achieved through the intermediary effect of residential satisfaction. Place attachment of young talent to cities cannot significantly regulate the impact of air pollution perception on residential satisfaction, but it can significantly regulate the relationship between residential satisfaction and urban settlement intentions. That is to say, although place attachment cannot reduce the decline in residential satisfaction brought by air pollution perception, it can weaken the negative impact of air pollution perception on dwelling willingness through a decline in residential satisfaction. This paper contributes to a deeper understanding of the relationship between air quality and young talent settlement intentions.

## 1. Introduction

In recent years, with the sustained and stable development of China’s economy and society, the attraction of China’s living environment to international immigrants has continued to increase. In 2020, the number of foreign persons living in the mainland reached 850,000, which rose from 600,000 in 2010, an increase of 42% (China National Bureau of Statistics, 2021). At the same time, along with a series of policies aimed at encouraging reasonable population migration, various traditional barriers that restrict population mobility have been broken down. Meanwhile, intensive regional competition has made attracting and retaining young talent an important approach to obtaining development advantages. The urban settlement of young talent has been an important issue in China. For example, in recent years, large- and medium-sized cities, such as Hangzhou, Nanjing, Wuhan, Zhengzhou, Chengdu, and Xi’an, have introduced “New Deals” for young talent, beginning a “talent grabbing” war. Retaining young talent for long-term residencies and, thus, developing cities has become a common concern for city administration and demographic researchers. Yet, young talent endures unstable career developments and a lack of social integration, owing to their short stays in new cities. This results in their increased willingness to move among cities, which makes urban settlement challenging [1]. The theory of planned behavior (TPB) suggests that psychological factors (such as attitude, subjective norms, and perceptual behavioral control) indirectly influence individual behavior through behavioral intentions [2]. Based on the TPB, scholars such as Weng and McElroy, proposed the concept of talent urban settlement intentions [3]; that is, the willingness of talent to work and live in a city for a long time. From this concept we can regard “settlement intentions” as a kind of attitude and behavior tendency, which reflects a comprehensive evaluation of talent from various aspects of a city and is a good indicator of a city’s ability to retain talent. A comprehensive analysis of the influencing factors of the willingness of talent to settle down can help city administration understand what such talent need from the city and thus enact targeted and efficient public policies and provide public goods.

Various studies have been conducted on urban settlement intentions and their influencing factors on population migration between countries, cities, and urban–rural areas. Related studies show that economic factors (e.g., income level and cost of living) [4], personal factors (e.g., education, occupation, and mobility distance) [5,6], amenities (e.g., entertainment and recreational facilities, cultural facilities, consumer shopping, green parks, and public cultural activities) [7], policy (e.g., household registration and subsidies) [8,9], and social factors (e.g., family and friends’ connections, traditional attitudes, and cultural inclusion) [10] are all important influencing factors in urban settlement intentions. In addition, when examining the influencing factors of overseas immigration, Poprawe believes that political corruption can easily lead people to emigrate abroad [11]; Bertoli and Moraga found that the migration situation between the two countries is affected, not only by various factors between the two countries, but also by the policies of a third country [12].

In recent years, as environmental hazards, e.g., air pollution, have become increasingly serious, more individuals consider air quality an important factor when choosing a place of residence (for physical and mental health purposes). A survey conducted by New Fortune magazine in 2013 found that the environment had become one of the important factors to promote to Chinese residents to consider international migration [13]. Based on city level data, Qin and Zhu’s research confirmed that, during a period of increasing air pollution, people retrieve “immigrants” through the Internet more frequently [14]. Kohlhuber et al. found that highly educated people are more concerned about air quality [15]. Jacquemin et al. further found that educational attainment was highly correlated with the level of perceived annoyance [16]; particularly, respondents with a graduate degree or higher were found to be the most sensitive to poor air quality. These findings suggest that young talent are more sensitive and concerned about air pollution, and that this may consequently influence their choice of cities when seeking long-term employment. However, existing studies related to population migration settlement have often neglected the topic of air quality [17]. Further, the impact mechanism of air pollution perception on urban settlement intentions has not been examined, although perception is considered important for initiating actions. With increasing concerns about air pollution, it is necessary to introduce air quality factors into analyses, focusing on their impact on the urban settlement intentions of young talent in China.

This paper is devoted to studying the impact of air pollution perception on the willingness of young talent to settle down in Chinese cities, based on the perspective of the inter-city talent competition using young talent as the research subjects. This study further shows how, and under what circumstances, air pollution perceptions affect the willingness of young talent to settle down in cities. Specifically, this study proposes urban residential satisfaction as a mediating variable and empirically tests its mediating role. It also suggests and tests the moderating role of place attachment in the relationship between air pollution perception and urban settlement intentions. The study systematically investigates the impact mechanisms and boundary conditions of air pollution perception on urban settlement intentions. It further provides theoretical references for city administration to improve the urban talent environment, urban talent attraction, and their retention. The study is structured as follows. Section 2 reviews existing studies and proposes hypotheses for testing. Section 3 introduces the research method and design used in this study. Section 4 presents the empirical results and conducts an in-depth discussion to interpret the results. Section 5 concludes this study by determining future studies.

## 2. Literature Review and Hypotheses Formulation

### 2.1. Impact of Air Pollution Perception on the Urban Settlement Intentions of Young Talent

By referring to perceived quality research, based on the consumer perspective [18], air pollution perception is defined as people’s opinion formed by the air pollution conditions around them and considers the processes by which the opinion is modified. Although there is no research that directly examines the impact of air pollution perception on the willingness of young talent to settle in cities, many scholars have conducted exploratory studies about the relationship between subjective and objective environmental quality and the migration and settlement of a population. Chen et al. analyzed the impact of air pollution on population migration in China from 1996 to 2010 [19] and found that, within the study period, air pollution reduced the in-migration of mobile populations by 50% in a county, ultimately reducing the total population by 5% through net out-migration. However, as Gronroos et al. [20] stated, customer satisfaction regarding a service stems from comparing the customer’s perception of a service and their expectations from the service. Accordingly, air pollution perception is a direct reflection of objective air quality, a complement of, and development in, objective air quality, and is a more direct influencing factor in decision-making. Air pollution perception may influence the willingness of young talent to settle down in cities in three ways. Concerns about their own and their family’s health may influence the willingness of young talent to settle down in cities. Air pollution can cause respiratory diseases, physiological dysfunctions, and irritate mucosal tissues, such as the eyes and nose, resulting in illnesses or the recurrence of old illnesses in people with a history of respiratory diseases [21]. Continuing to live in a polluted environment may place residents at an increased risk of contracting heart disease and lung cancer [22]. Sun et al. [17] analyzed data from a 2014 survey conducted by the National Health and Family Planning Commission in eight cities, including Beijing, Xiamen, and Shenzhen, and found that, as the concentration of PM2.5 increased, people’s health-related expectations decreased significantly. Consequently, young talent may choose to leave such cities. The willingness of young talent to settle in cities may also be influenced by air pollution surpassing all other environmental issues and increasingly affecting residents’ trust in the government, becoming a political, societal, and living condition issue [23]. Wang and Han [24] found that air pollution perception significantly affects the public’s evaluation of the government’s performance. They also found that poor air perception may cause young talent to lose trust in local governments, thus losing interest in settling down in a city. Moreover, additional living expenses for young talent caused by air pollution perception may also influence their willingness to settle down. Further, they are required to pay for the explicit costs of protective equipment, such as air purifiers, and shoulder various hidden costs, such as reduced labor efficiency [25] and increased workdays owing to air pollution. For example, studies show that a 1% increase in suspended particulate matter in the air is associated with a significant increase of 0.44% in the number of workdays lost [26]. The combination of these costs is not negligible and constitutes a “push” factor in migrating from cities.

Therefore, to further explore the relationship between air pollution perception and the settlement intentions of young talent, this study proposes the following hypothesis based on the abovementioned analysis:

**Hypothesis** **1** **(H1).***Air pollution perception has a significant negative effect on the urban settlement intentions of young talent*.

### 2.2. The Mediating Role of Residential Satisfaction between Air Pollution Perception and the Urban Settlement Intentions of Young Talent

#### 2.2.1. Air Pollution Perception and Urban Residential Satisfaction

Satisfaction occurs when people’s inner desires and all their subjective feelings are in tune with each other. It is reflected in a person’s psychological state, when one is extremely pleased and comfortable [27]. Scholars, such as You and Chen [28], defined urban residential satisfaction as the public’s satisfaction assessment, based on the accumulation of all environmental feelings. An empirical study on the elderly found that subjective evaluations, that is, perceived factors, explained the degree of variation in residential satisfaction much more effectively than objective environmental variables [29]. Objectively measured air quality ultimately affects residential satisfaction by influencing the subjective perceptions of young talent. Air pollution directly triggers sensory discomfort, which consequently causes negative emotions. Moreover, studies show that, although individuals have many behavioral beliefs, only a relatively small number of these are accessible at any given time and in any given context. These obtainable beliefs, also referred to as salient beliefs, are the cognitive and emotional bases for behavioral attitudes, subjective norms, and perceptual behavioral control [30]. Young talent is more concerned about air quality and have higher expectations regarding this issue [31]. Therefore, for young talent, air pollution perception becomes a salient belief that profoundly influences residential satisfaction. If their perception of air quality differs from their preferred conditions, it causes a negative assessment of their satisfaction with urban living. Therefore, perceived air pollution is a subjective perception and an important aspect of the urban environment from the point of view of young talent and is likely to significantly influence residential satisfaction.

Accordingly, the study hypothesizes the following:

**Hypothesis** **2a** **(H2a).***Air pollution perception has a significant negative impact on the urban residential satisfaction of young talent*.

#### 2.2.2. Residential Satisfaction and the Urban Settlement Intention of Young Talent

People tend to follow practices that satisfy them. In other words, satisfaction drives loyalty. Cardozo [32] suggested that customers are more likely to purchase a product again (or other goods) if they are satisfied with a merchant’s production. Similarly, if young talent has high-level residential satisfaction in a city, they are more likely to continue living there. In other words, residential satisfaction generates a willingness to settle down in cities. Weng et al. [33] proposed the concept of regional commitment in his study on talent agglomeration, arguing that, if the regional environment matches the growth and living needs of talent, that is, if they experience good residential satisfaction, it can generate regional commitment between the talent and the city. This will, in turn, result in the desire to actively work to promote regional development. Therefore, residential satisfaction evaluates the degree to which the needs of young talent are met. According to customer loyalty and regional commitment theories, high-level urban residential satisfaction can make young talent emotionally “loyal” to a city, promote strong regional commitments, and make them willing to invest more in regional development. Liang [34] found that satisfaction with urban life has a significant impact on the willingness of a migrant population to settle down, especially those who are “relatively satisfied” and “very satisfied”. Yao [35] conducted a large-scale questionnaire survey of foreign residents in Shanghai and found that the higher the satisfaction of foreign migrants with the city and community, the greater the likelihood of their long-term residence in Shanghai.

Accordingly, this study hypothesizes the following:

**Hypothesis** **2b** **(H2b).***Residential satisfaction has a significant positive impact on the urban settlement intentions of young talent*.

#### 2.2.3. Residential Satisfaction Plays an Intermediary Role in the Impact of Air Pollution Perception on the Urban Settlement Intentions of Young Talent

Residential satisfaction reflects whether residents’ various expectations and needs from a city are met, resulting in a pleasant and positive state of mind [36]. It essentially reflects the comparison of people’s perceptions with their expectations in the context of various urban residence-related factors and satisfaction arises when perceptions meet expectations [37]. Young talent is a group of urban residents who are significantly concerned with environmental issues and the impact of air quality factors on health [31]. If their perception of air quality is not satisfactory, they will feel dissatisfied and choose to relocate to a better environment. Previous research further shows that residential satisfaction is a positive psychological mechanism that can explain proactive psychological processes, such as urban settlement intentions. Further, it links the perceived environment with the intentions of long-term residence [38]. Therefore, this study argues that, for young talent, residential satisfaction may be a psychological mechanism that can effectively convey perceived environmental quality as a livability characteristic; that is, the perception by young talent of air pollution is influenced by the psychological mechanisms of residential satisfaction. This further impacts their willingness to stay in a new city.

This study therefore combines the Hypotheses H2a and H2b and proposes the following hypothesis:

**Hypothesis** **2** **(H2).***Residential satisfaction plays an intermediary role in the relationship between air pollution perceptions and the urban settlement intentions of young talent*.

In other words, air pollution perceptions influence the urban settlement intentions of young talent through residential satisfaction.

#### 2.2.4. The Moderating Role of Place Attachment in the Relationship between Residential Satisfaction and Urban Settlement Intentions

Place attachment is a concept in psychology that characterizes the emotional bond and psychological identity between an individual and a specific environment. It refers to an individual’s emotional response to their interaction with the environment and reflects a deep emotional connection to the place. Cultural and social characteristics modify this human–place relationship [39]. Psychological attachments to a place, or place attachment, occurs when individuals assign specific values to places in human–place interactions and form a positive emotional tie [40,41]. Place attachment may arise from cognition (i.e., the more you know about a place, the more you love it), well-known or reciprocal social networks, or a special emotional connection [42]. Cassn et al. [43] suggest that people’s attitudes and behaviors toward a particular place are significantly influenced by the emotions, meanings, and values that they assign to that place. Young talent, after working and living in a city, become inextricably linked to the city in various ways. A positive link generates positive emotions and is assigned special values, thus developing place attachment. Greater place attachment means that they have special emotional and social ties to a city and a deeper sense of identity and belonging. Therefore, they are likely to give better satisfaction ratings, even when they perceive air quality as being poor.

Therefore, this paper hypothesizes the following:

**Hypothesis** **3a** **(H3a).***Place attachment inversely regulates the relationship between air pollution perception and urban settlement intentions*.

In other words, relative to weaker place attachment, stronger place attachment will weaken the impact of poor air pollution perceptions on residential satisfaction.

Further, place attachment may not only affect the relationship between perceived air pollution and attitude, but it may also influence the relationship between attitude and behavioral intentions. Although there is still no research in this area, studies in other fields have found similar findings. Scholars studying the loyalty of tourists to tourist destinations show that place attachment significantly influences the relationship between tourist perceptions of a destination and satisfaction with tourism, as well as loyalty [44]. Li and Zhou [45] found that place attachment, as a moderating variable, significantly reinforced positive behavior among tourists, such as protecting tourist attraction sites. Therefore, in conjunction with the analysis above, it can be hypothesized that this moderating factor may exist between residential satisfaction and the urban settlement intentions of young talent. The inclination of young talent to leave a city owing to lower residential satisfaction arising from what they perceive to be poor air quality may be weakened when there is good place attachment. Therefore, this study suggests that place attachment may have a moderating effect on the relationship between residential satisfaction and the willingness to settle down. Accordingly, this study proposes the following hypothesis:

**Hypothesis** **3b** **(H3b).***Place attachment plays a moderating role in the relationship between residential satisfaction and the urban settlement intentions of young talent and enhances the positive impact of residential satisfaction on the willingness to settle down*.

Based on the above analysis, this study argues that place attachment between young talent and a city creates a deep emotional connection and a special value ascription between them and the environment. This consequently changes their satisfaction assessment of the air quality of the living environment and the resulting urban settlement intentions.

Combining H3a and H3b, this study proposes the following hypothesis:

**Hypothesis** **3** **(H3).***Place attachment plays a moderating role among the relationship of air pollution perception, residential satisfaction, and urban settlement intentions; despite a greater place attachment, young talent may still suffer lower residential satisfaction owing to poor air quality. However, the moderating role of place attachment weakens the impact of residential satisfaction on their urban settlement intentions*.

The proposed hypotheses can be further summarized, as in Figure 1.

## 3. Materials and Methods

### 3.1. Data Collection

This study took Hangzhou as the research area. Hangzhou is the capital city of Zhejiang Province, located in the south of China. The city covers a total area of 16,850 square kilometers and the local GDP is 1.61 trillion CNY (Chinese Yuan, namely 0.23 trillion US dollars) with a residential population of 11.936 million in 2020 (Hangzhou Municipal Bureau of Statistics, 2021, the data mentioned below in this paragraph area also from this). Hangzhou is devoted to developing a digital economy and achieving high-quality development, which needs a large amount of young talent. By the end of 2020, Hangzhou’s talent pool expanded to 2.945 million people with an annual increase of 5.2%. The net inflow rate of talent and overseas talent ranked at the top in China, and, for 10 consecutive years, Hangzhou has been rated as being among the “Ten Most Attractive Chinese Cities for Foreigners” (Hangzhou Municipal Bureau of Statistics, 2021). Hangzhou is becoming one of the most dynamic cities in China and is therefore suitable for investigating the young talent settlement intentions and the influence of air pollution perceptions [46].

According to the World Health Organization, People aged 14 to 44 are classified as young people. So, in this study, “young talent” refer to those who have a Junior college degree or above and are under the age of 44. This study uses data gathered from a questionnaire survey conducted by the research team, from April to June 2018, at the Hangzhou Future Sci-Tech City, the Hangzhou Economic and Technological Development Zone, Hangzhou High-Tech Zone (Binjiang), and key office buildings in central urban districts, where the inflow of young talent to Hangzhou is more concentrated. The target population of the questionnaire survey was young talent working in Hangzhou. The survey adopted a stratified random sampling method. With support from relevant government departments, survey respondents were randomly selected from a list of enterprises and their employees. A total of 1200 questionnaires were distributed with 300 for each abovementioned region. Overall, 1089 responses were collected, and, of these, 102 were excluded due to incomplete or invalid data. The final valid number of questionnaire responses was 987. Table 1 provides basic information about the survey respondents. A total of 807 (81.8% of the total sample) had a bachelor’s degree or above. This included 49 (5%) with a doctoral degree and 176 (17.8%) with a master’s degree. Further, 702 (71.1%) had an annual income of CNY 80,000 or more and 761 (77.1%) were under 35 years old. Based on age, educational attainment, and income level, the survey sample mainly consisted of young people with high educational attainments.

### 3.2. Measurement

The questionnaire’s measurement scales were based on those used by previous related studies. A small-scale pre-survey and analysis were conducted before the large-scale survey. During this process, the research team communicated with the respondents comprehensively and used their input to improve the questionnaire in terms of reliability, validity, readability, and semantic accuracy to avoid any possible ambiguities arising from terminology. Chinese was used as the common language and due attention was also paid to minimize information loss during translation in paper writing [47]. The questionnaire items used a five-point Likert scale, and each respondent answered based on their judgment. In the questionnaire, an answer of 0 corresponded to “no such problem”, 1 to “average”, 2 to “not serious”, 3 to “not very serious”, 4 to “quite serious”, and 5 to “very serious”.

Based on Li [48] and Wang and Han [24], this study measured air pollution perception using three items, “the severity of PM2.5 in Hangzhou”, “the air in Hangzhou is gray”, and “the air is smelly in Hangzhou”. To evaluate place attachment, this study referenced the studies by Kyle et al. [49] and Williams and Vaske [50]. Seven items are used to measure place attachment, “I like the cultural heritage of Hangzhou”, “the landscape of Hangzhou gives me a sense of belonging”, “Hangzhou possesses all kinds of living facilities that I want”, “I have a good time with my colleagues (neighbors) in Hangzhou”, “the help provided by the people around me makes me feel very warm”, “I often feel respected in my life”, and “I am willing to make efforts to make Hangzhou become better”. To evaluate urban settlement intentions, the scale, based on the settlement intention scale developed by Hu and Weng [51] was adjusted so that it included four questionnaire items: “I am willing to stay and live in Hangzhou for a long time”, “I have not considered the idea of settling in other similar cities”, “if I were to choose again, I would still choose to work and live in Hangzhou”, and “if I have the opportunity, I would recommend my relatives and friends from other places to live in Hangzhou”. Satisfaction of living in a city was evaluated using three items, “I am happy to be able to work and live in this area”, “I am satisfied with the living environment in the city”, and “I often feel spiritually happy living here”. Simultaneously, drawing on the results from existing research, this study selected seven control variables: gender, age, education, time spent in Hangzhou, income, development expectations, and family and friends in Hangzhou. Among them, gender and education were dummy variables. Variable “1” represented male in the gender variable and having at least one family member or friend in Hangzhou in the family and friends variable. Regarding the education variable, an educational attainment of junior college was considered a reference point. Other variables, such as age, time spent in Hangzhou, income, and career development expectations, were considered as continuous variables.

## 4. Results

### 4.1. Reliability and Validity Tests of the Questionnaire

This study used Cronbach’s alpha coefficient to test the reliability of the measurement’s variables to ensure the reliability and validity of the questionnaire. The results showed that the reliability of the four scales of place attachment, residential satisfaction, air pollution perception, and urban settlement intentions were 0.868, 0.974, 0.912, and 0.934, respectively. All four results were greater than 0.7 and, thus, had good reliability [52]. Further, according to the method suggested by Fornell et al. [53], this study used AMOS 24.0 (software to analyze structural equation modeling) to conduct a confirmatory factor analysis (CFA) on the four main variables to calculate the square root of the average variance extracted (AVE) value of each variable. The discriminant validity of each variable was tested by comparing the square root of the AVE value of each variable with the correlation coefficient between the latent variables, as shown in Table 2. The square root of the AVE value of all the variables was greater than the correlation coefficients. This indicated a good discriminant validity among the variables.

In addition, this study conducted a structural validity test on the four variables. The results showed that all the factor loading values in the four-factor model (model fit indices: *X*^2^/df = 4.684, RMSEA = 0.078, IFI = 0.925, CFI = 0.918) (*X*^2^ denotes chi-square test, which can assess overall fit and the discrepancy between the sample and fitted covariance matrices. df denotes model degrees of freedom. The chi-square value and model degrees of freedom can be used to calculate a *p*-value. Model is good fit if *p*-value > 0.05. RMSEA is an abbreviation for Root mean Square Error of Approximation. It is a parsimony-adjusted index, which is good fit if RMSEA < 0.08. IFI is an abbreviation for incremental fit index with values greater than approximately 0.90. CFI is an abbreviation for comparative fit index, which is good fit if CFI ≥ 0.90) were significantly higher than the general recommendation of 0.4, indicating that the measurement items of each variable could be better aggregated and effectively reflect the same construct. The results also showed that the four-factor model substantially fit indicators better than the other factor models. In summary, the tests described above indicate that the data of this questionnaire have high reliability and validity.

### 4.2. Testing the Main Effect

Models 1 and 2 in Table 3 show that, after controlling the relevant variables, the independent variable of air pollution perception significantly impacts the dependent variable of urban settlement intentions. The *R*^2^ changes significantly, supporting H1 (β = −0.077, *p* < 0.01). Particularly, it is worth pointing out that, among the control variables, family and friends in Hangzhou and career development expectations have a significant impact on urban settlement intentions. In other words, migrant talent with family and friends in Hangzhou are more willing to settle in Hangzhou permanently. Additionally, development expectations are also a key influencing variable of urban settlement intentions. Development expectations depend on one’s judgment of future employment and development prospects; the better the expectation, the greater the cost of “giving up.” In 2019, the added value of Hangzhou’s core digital economy industry was CNY 379.5 billion, which increased by 15.1% compared with 2018. Contrastingly, the growth was 14.6% for the e-commerce industry, 13.6% for the Internet-of-Things industry, and 15.7% for the software and information service industry. Such growths resulted from the rapid development of high-tech industries, which provide a career platform for young talent and raises their development expectations. These improvements made Hangzhou one of the top cities in China, in terms of young talent inflow.

### 4.3. Testing the Mediating Effects

Models 3–6 tested the mediating effects of air pollution perception on residential satisfaction and further influence on the urban settlement intentions of young talent. Firstly, according to the results of Models 3 and 4 in Table 3, the control variables, such as age and career development expectations, had a significant influence on residential satisfaction and remained robust both in Models 3 and 4. After the inclusion of the key independent variables air pollution perception in Model 4, its effect on residential satisfaction was significant (β = −0.167, *p* < 0.001), with *R*^2^ changing to 0.024, further enhancing the model’s explanatory power. Therefore, Hypothesis H2a was supported. Further, intermediary variables were included in Model 5. The results showed that residential satisfaction had a significant influence on the urban settlement intentions of young talent (β = 0.530, *p* < 0.001). The model’s explanatory power increased by 23.3%, based on the amount of change in *R*^2^; thus, supporting Hypothesis H2b. In addition, this study further examined the effect of perceived air quality and residential satisfaction on settlement intentions. Model 6 incorporated both the independent variable of air pollution perception and the mediating variable of residential satisfaction. The empirical results showed that residential satisfaction had a significant influence on settlement intentions (β = 0.532, *p* < 0.001), while the influence of air pollution perception became insignificant (β = −0.012, *p* > 0.05). After adding both the mediating and independent variables, the independent variable in the model became insignificant. Contrastingly, the mediating variable remained significant, according to the evaluation method of Baron and Kenny [54]. This indicates that residential satisfaction plays a mediating role between air pollution perception and urban settlement intentions altogether, thus verifying Hypothesis H2.

To further test the mediating effects, this study conducted a bootstrap test using the PROCESS macro for SPSS/SAS developed by Hayes [55], and repeated the sample 5000 times. The results showed that the indirect effect of air pollution perception on urban settlement intentions through residential satisfaction was 0.0886, with a 95% confidence interval of [0.055, 0.124] and *p* < 0.001. According to the criteria proposed by Preacher and Hayes [56] for testing mediating effects, if the confidence interval of the indirect effect does not include 0, then the indirect effect reaches a significant level. The empirical results show that the exclusion of a value of 0 also confirmed Hypothesis H2.

### 4.4. Testing the Moderating Effects

Hypothesis H3a proposes that place attachment has a positive moderating effect on the relationship between air pollution perception and residential satisfaction. In other words, greater place attachment can weaken the effect of air pollution perception on residential satisfaction and offset the decrease in residential satisfaction caused by poor air quality. To test this hypothesis, residential satisfaction was set as the dependent variable per the three-step test method of moderated hierarchical regression analyses. Hierarchical regression was established to sequentially incorporate the control variable, standardized independent variable, moderating variable, the product of the moderating variable, and independent variable into the equation, as shown in Models 7 and 8 in Table 4. The interaction term insignificantly influenced residential satisfaction (β = 0.004, *p* > 0.05). Therefore, Hypothesis H3a was not verified. Further, this study used the same approach to test the moderating role of place attachment between residential satisfaction and urban settlement intentions. Models 9 and 10 showed the empirical results, where the product term had a significant influence on settlement intentions, with β = 0.024 (*p* < 0.01) and Δ*R*^2^ = 0.132 (*p* < 0.05). Therefore, Hypothesis H3b was verified; in other words, the relationship between residential satisfaction and urban settlement intentions is significantly stronger when place attachment is stronger.

To visualize the moderating role played by place attachment in the relationship between residential satisfaction and settlement intentions, this study plotted the moderating relationship based on the method recommended by Aiken and West [57]. Each chosen variable had one standard deviation above and below the mean. Further, the moderating relationship is plotted in Figure 2, and shows the difference in the relationship between residential satisfaction and settlement intentions when young talent has different levels of place attachment to the city. With greater place attachment, a slight change in residential satisfaction promotes an increase in urban settlement intentions. Conversely, a weaker place attachment weakens the positive impact of residential satisfaction on urban settlement intentions.

Moreover, this study uses the PROCESS plug-in for SPSS/SAS developed by Hayes to further test the adjusted mediating effect by repeating the sample 5000 times using the bootstrap method. The empirical results show that the mediating effect of residential satisfaction between air pollution perceptions and urban settlement intentions differs significantly at different levels of place attachment. Air pollution perceptions had a stronger indirect negative effect on urban settlement intentions at lower levels of place attachment (β = −0.071, *p* < 0.001), whereas at higher levels of place attachment, it had a weaker negative effect on settlement intentions (β = −0.038, *p* < 0.001), (Δβ = −0.033, *p* < 0 05), with a 95% confidence interval of [−0.021, −0.001], which does not include 0. Therefore, Hypothesis H3 was also empirically supported. With a greater level of place attachment, young talent may still experience residential dissatisfaction due to poor air quality. However, the impact of residential dissatisfaction on urban settlement intentions is weakened by greater place attachment.

## 5. Discussion

Based on data obtained from 987 questionnaires collected from a sample group of young talent in Hangzhou, this study explored the influence of perceived air quality on urban settlement intentions. Hypotheses 1, 2, 2a, 2b, 3 and 3b were supported, whereas Hypothesis 3a was rejected. The findings of this study are as follows. First, air pollution perceptions significantly influence the urban settlement intentions of young talent; the poorer their air pollution perception, the weaker the urban settlement intentions. Second, residential satisfaction significantly mediates the relationship between air pollution perception and urban settlement intentions. In other words, a poorer air quality perception reduces residential satisfaction among young talent, thus weakening their urban settlement intentions. In conclusion, place attachment has a significant moderating effect on the relationship between residential satisfaction and settlement intentions. However, it insignificantly affects the relationship between air pollution perception and residential satisfaction. Compared with a weaker level of place attachment, a greater level of place attachment does not change the dissatisfaction associated with poorer perceived air quality. Instead, it can weaken the effect that this dissatisfaction has on settlement intentions.

Given the current situation of talent competition among various cities, this study focuses on the urban settlement intentions of young talent at an earlier stage. For the first time, it introduces the perceived air quality factor into the analysis of the influencing factors of the urban settlement intentions of young talent. This study examined the influence mechanism of air pollution perception on the urban settlement intentions of young talent. This study also verified the applicability of the demographic characteristics, economic development, and socio-cultural factors proposed by Woon et al. [4], which influence population migration, on young talent. The idea that young and highly educated people are more concerned about air pollution, as suggested by Jacquemin et al. [16], was expanded and confirmed, further corroborating the idea that young talent also choose to “vote with their feet” in the face of environmental pollution, as demonstrated by Banzhaf and Walsh [58]. In addition, unlike in previous studies, which used objective indicators to examine the impact of specific air pollutants on population migration and settlement, this study explored the psychological mechanism of the impact of air quality on the settlement intentions of young talent, starting from the concept of perceived air quality and with the help of factors, such as residential satisfaction and place attachment. This line of exploration is a useful supplement to studies on the impact of objective air quality on the spatial mobility of labor forces. Furthermore, based on the TPB, this study explored, for the first time, the influence mechanism of the relationship between perceived air quality and the settlement intentions of young talent. It was found that residential satisfaction mediates the relationship altogether, thus uncovering the transmission mechanism from better and worse perceived air quality to the strength of urban settlement intentions. The study demonstrated that air pollution perception significantly influences the satisfaction of young talent with urban living, affecting their settlement intentions. Urban air quality should also become an important aspect of the urban talent environment and be incorporated in the government’s public service provisions; thus, the study provides a reference for subsequent studies related to talent environment construction and evaluation. Fourth, this study tested the moderating role of place attachment, an important concept in geography that refers to the emotional interaction between people and specific places, in the relationship among the independent (air pollution perception), mediating (residential satisfaction), and dependent (settlement intention) variables. The findings of this study expand Li and Zhou’s research [45] on the moderating role of place attachment. Additionally, based on the results of the empirical analysis, it was verified that, in terms of human–place interaction, place attachment does not have a moderating effect on the relationship between air pollution perception and residential satisfaction. Place attachment, however, has a moderating effect on the impact of residential satisfaction on settlement intentions. Therefore, this study explained, in more detail, the influence mechanism of air pollution perception on the urban settlement intentions of young talent.

## 6. Conclusions

This study empirically examined the significant influence of air pollution perception on the urban settlement intentions of young talent and explained its influence mechanism. In the context of fierce competition for talent across various cities, this study’s findings serve as an important reference for cities to construct an advantageous ecological environment for their talent and enhance the city’s competitiveness for young talent. First, various cities are currently attracting talent through different policies, such as relaxing household registration restrictions and providing subsidies. However, the influence of environmental factors, such as air pollution on young talent, should not be ignored. Air pollution management should be enhanced through the strategic construction of an urban talent environment to promote high-quality development. Air quality should further be improved continuously through measures such as adjusting industrial structures and strengthening pollution control and dust management in key industries. Second, as improving air pollution control involves complex factors, such as industrial transformation and upgrades and synergy between multiple locations, it must be a long-term process. Greater place attachment can offset the reduction of urban settlement intentions caused by low residential satisfaction due to air pollution. Based on relevant sources, city managers can enhance place attachment to their respective cities by strengthening and improving publicity, creating a compassionate and welcoming image, building community exchange platforms to facilitate social integration, creating opportunities to attract young talent to participate in urban governance, and increasing the understanding and identity of young talent within the city. In conclusion, guided by the various housing needs of young talent, it is necessary to improve the supply of various public goods in the city, enhance urban governance, and implement multiple measures to improve residential satisfaction in the city.

This study, however, has some limitations. First, the survey data were obtained from Hangzhou. Although the inflow of young talent to Hangzhou in recent years has been among the highest in China and the survey sample shows that the sources of talent are also distributed across the country, the survey is inevitably influenced by the geographical characteristics of Hangzhou. Surveys and studies with greater coverage are yet to be conducted to verify whether the research findings can be generalized to the national level. Second, while semi-structured interviews were conducted with young talent, the formal survey was conducted in a relatively short period, making it difficult to properly reflect the dynamic interactions among variables, such as air pollution perception, residential satisfaction, place attachment, and settlement intentions. Follow-up studies can adopt a longitudinal tracking approach to conduct an extensive analysis of the relationship among these variables. This approach would improve the persuasiveness of the research findings. In conclusion, regarding the influence mechanism of air pollution perception on urban settlement intention, whether other variables can be included in the research model should be further explored in future studies.

## Figures and Tables

**Figure 1 ijerph-19-01080-f001:**
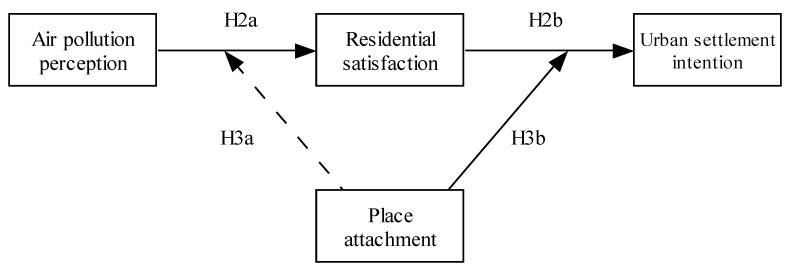
The conceptual model.

**Figure 2 ijerph-19-01080-f002:**
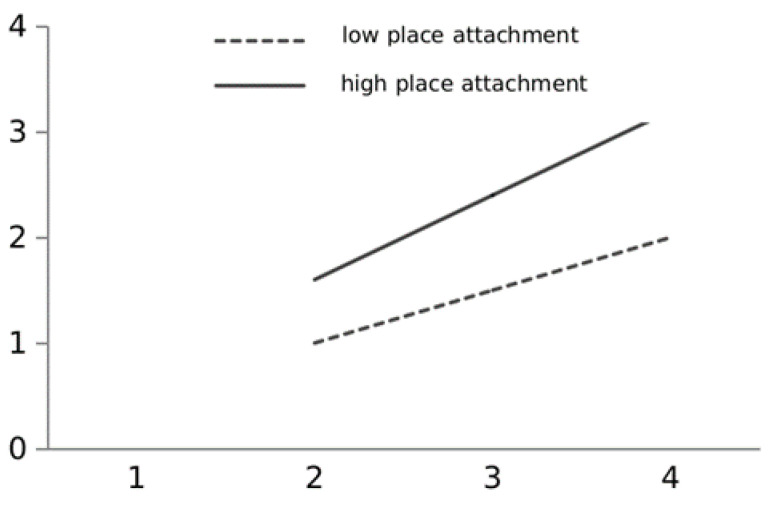
The regulatory role of place attachment on the relationship between residential satisfaction and urban settlement intention.

**Table 1 ijerph-19-01080-t001:** Sample of the respondents.

Statistical Indicators	Ratio
Gender	
Male	55.5%
Female	44.5%
Age (years)	
Below 22	2.3%
Between 22 and 27	32.1%
Between 27 and 35	42.7%
Above 35	22.9%
Education	
Junior college	18.2%
Bachelor	59.0%
Master	17.8%
Ph.D.	5.0%
Hometown	
Hangzhou	19.8%
Outside Hangzhou but within Zhejiang	34.2%
Outside Zhejiang or China	46.0%
City of final graduation	
Zhejiang	48.6%
Outside Zhejiang	51.4%
Annual income (CNY)	
Below 80,000	28.9%
Between 80,000 and 150,000	38.2%
Between 150,000 and 250,000	18.2%
Between 250,000 and 350,000	7.6%
Above 350,000	7.1%

**Table 2 ijerph-19-01080-t002:** Reliability and validity tests.

Variable	Mean	S.D.	1	2	3	4
Air pollution perception	2.562	1.009	**0.690**			
Place attachment	3.773	0.625	0.616 ***	**0.800**		
Residential satisfaction	3.821	0.759	0.218 ***	0.284 ***	**0.075**	
Urban settlement intention	3.800	0.721	0.451 ***	0.437 ***	0.511 ***	**0.780**

Note: The numbers in bold form a diagonal, and the diagonal line demonstrates the square root of the average variance extracted (AVE) value, and below the diagonal is the correlation coefficient of each variable. *** *p* < 0.01 with two-tailed test.

**Table 3 ijerph-19-01080-t003:** Testing main effect and mediating effects of residential satisfaction.

Variable	Urban Settlement Intention		Residential Satisfaction		Urban Settlement Intention	
	Model 1	Model 2	Model 3	Model 4	Model 5	Model 6
Key variable						
Air pollution perception		−0.077 ** (0.031)		−0.167 *** (0.031)		−0.012 (0.027)
Mediating variable						
Residential satisfaction					0.530 *** (0.027)	0.532 *** (0.027)
Control variable						
Gender	−0.137 ** (0.061)	−0.139 ** (0.061)	−0.166 (0.062)	−0.169 ** (0.061)	−0.049 (0.052)	−0.049 (0.052)
Age	0.091 * (0.050)	0.090 ** (0.050)	0.084 * (0.050)	0.082 * (0.050)	0.046 (0.042)	0.045 (0.038)
Edu1	−0.137 * (0.179)	−0.143 * (0.079)	−0.001 (0.080)	−0.013 (0.079)	−0.137 ** (0.066)	−0.136 (0.067)
Edu2	−0.112 (0.103)	−0.116 (0.103)	−0.021 (0.104)	−0.029 (0.103)	−0.101 (0.087)	0.100 (0.087)
Edu3	0.047 (0.160)	0.041 (0.160)	0.090 (0.162)	0.079 (0.060)	−0.001 (0.035)	0.001 (0.135)
Time spent in Hangzhou	0.033 (0.035)	0.030 ** (0.035)	0.006 (0.035)	0.013 (0.035)	0.037 ** (0.029)	0.037 ** (0.029)
Annual income	0.029 (0.031)	0.032 (0.031)	0.034 (0.032)	0.041 (0.031)	0.011 (0.026)	0.010 (0.026)
Career development expectations	0.465 *** (0.032)	0.433 *** (0.034)	0.432 *** (0.032)	0.363 *** (0.034)	0.236 *** (0.029)	0.240 *** (0.031)
Family and friends	0.433 ** (0.090)	0.430 *** (0.085)	0.136 ** (0.043)	0.113 ** (0.087)	0.237 ** (0.079)	0.337 ** (0.093)
Constant	−1.808 *** (0.170)	−1.696 *** (0.175)	−1.875 *** (0.172)	−1.632 *** (0.176)	−0.815 ** (0.152)	−0.828 (0.155)
*R* ^2^	0.188	0.193	0.166	0.190	0.421	0.422
Adjusted *R*^2^	0.181	0.185	0.159	0.183	0.416	0.414
Δ*R*^2^	--	0.05 ***	--	0.024 ***	0.233 ***	0.001 ***
*F*	28.274 ***	25.948 ***	24.387 ***	25.485 ***	359.169 ***	10.200 *
VIF	1.880	1.879	1.879	1.880	1.879	1.880

Note: Standard errors in parentheses; * *p* < 0.10, ** *p* < 0.05, *** *p* < 0.01 with two-tailed test; all the regression coefficients were non-standardized. VIF, Variance Inflation Factor.

**Table 4 ijerph-19-01080-t004:** Testing the moderating effects of place attachment.

Variable	Residential Satisfaction		Urban Settlement Intention	
	Model 7	Model 8	Model 9	Model 10
Key variable	−0.078 ** (0.033)	−0.077 ** (0.033)		
Air pollution perception	−0.078 ** (0.033)	−0.077 ** (0.033)		
Moderating variable				
Place attachment	0.271 *** (0.040)	0.273 *** (0.042)	0.114 *** (0.023)	0.123 *** (0.024)
Mediating variable				
Residential satisfaction			0.357 *** (0.020)	0.361 *** (0.020)
Interaction		0.004 (0.025)		0.024 ** (0.013)
Control variable				
Gender	−0.148 ** (0.059)	−0.113 ** (0.045)	−0.032 (0.037)	−0.033 (0.037)
Age	0.050 (0.049)	0.038 (0.037)	0.021 (0.030)	0.020 (0.033)
Edu1	−0.048 (0.077)	0.036 (0.059)	0.116 ** (0.047)	0.117 ** (0.047)
Edu2	−0.062 (0.101)	0.047 (0.076)	0.041 (0.097)	0.092 * (0.062)
Edu3	0.014 (0.040)	0.013 (0.119)	0.089 (0.062)	0.045 (0.097)
Time spent in Hangzhou	0.003 (0.034)	0.002 (0.026)	0.029 * (0.021)	0.028 * (0.021)
Annual income	0.051 * (0.031)	0.039 * (0.023)	0.014 (0.019)	0.014 (0.019)
Career development expectations	0.214 *** (0.040)	0.213 *** (0.040)	0.103 *** (0.025)	0.100 *** (0.025)
Family and friends	0.211 ** (0.097)	0.330 ** (0.091)	0.235 ** (0.089)	0.326 ** (0.063)
Constant	0.949 (0.198)	−0.950 *** (0.199)	3.513 *** (0.124)	3.516 *** (0.123)
*R* ^2^	0.227	0.227	0.306	0.438
Adjusted *R*^2^	0.219	0.219	0.301	0.431
Δ*R*^2^	0.061	0.00	0.118	0.132
*F*	28.717 ***	26.082	75.341 ***	68.948 ***
VIF	1.981	2.227	1.885	1.888

Note: Standard errors in parentheses; * *p* < 0.10, ** *p* < 0.05, *** *p* < 0.01 with two-tailed test; all the regression coefficients were non-standardized.

## Data Availability

Data will be made available upon request.

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
