# Peer review of "The Impact of Air Pollution Perception on Urban Settlement Intentions of Young Talent in China"

_ijerph, 2022, doi:10.3390/ijerph19031080_

Round 1

Reviewer 1 Report

The paper is a good outcome of an interesting topic for planners, decision-makers, and major relevant stakeholders. Also, the work is a timely one for international scholarly communities. However, I have a few comments to improve the paper as follows:

  1. You have hit the issue in the first line of your introduction. This issue is not existing in China only. We are familiar to see the similar issues in New Delhi for air pollution, Vancouver for housing crisis, and similar issues in California. I would suggest that the authors highlight the talent movement across the world in the light of several factors and bring the interesting issue of air pollution in the second paragraph. This will certainly enhance the visibility of the timely work to other scholars.
  2. line 7, this may be the first initiative in scientific article but the issue maybe highlighted in several other sources. So I suggest to revise the sentence
  3. Literature looks good with the hypothesis building. 
  4. Line 271, you must cite the appropriate source of 5.2% increase
  5. results look interesting in table 3, 4 with the model outcomes.
  6. The discussion looks good. However, I would like the authors to keep the discussion under a single headline. Open up another section as concluding remarks to mention about your work and its novelty, future scopes of the work to introduce similar methods to other parts of the world, and conclude your thoughts about the outcome.

Reviewer 2 Report

The topic of the article is relevant to the journal and promises more clarity to a somewhat fuzzy discussion about the impact of air pollution on individual preference to settle in urban areas. The structure of the article is straightforward and well-articulated. After the introduction section, the literature review and hypothesis formulation assess the literature and hypothesis for upcoming analysis. This is followed by the material and methods section reflecting the data collection process and measurements used. The results section provides the empirical insights and reflects the results based on the hypothesis discussed in the previous section. The outcomes are further discussed in the next section, including conclusions and outlook. The sequence of sections is logical, and the overall article structure is straightforward. 

Methodologically, the approach taken by the authors appears to be sound. However, the article needs attention to address a few minor issues that are hindering the authors' objectives.

  • Linguistic quality needs to be improved (several grammatical/syntactical errors throughout) ex: line 25.
  • Avoid using very complex statements where possible.
  • The authors presented the study area as international in the paper, but the survey was conducted in Chinese. The author could consider commenting on this contradictory aspect. 
  • The result section referred to some statistical aspects without referring to the particular model they discussed in the text. This makes it hard for the reader to follow the text and the tables. Please incorporate further details to make it more readable. 
  • The results of the Bootstrapping tests are not well defined in the tables and come in the middle of the results. I suggest authors also include them and then refer to the text for consistency with text flow and its implications for the results.  

Overall, the paper’s scientific contribution is of great value to future planners and governance perspectives for considering air pollution as a crucial aspect.

Reviewer 3 Report

Authors say (line 77-79)  “This study is the first to explore the impact of air pollution perception on the willingness of young talents to settle down in Chinese cities, based on the perspective of the intercity talent competition while using young talents as the research subject”.

  1. I think authors have to better define what is “young talent” because in Table 1 I can see that there are respondents with age above 35 and with education as Junior college.
  2. How young talents are selected?
  3. Also a definition of (keyword) “urban settlement intentions” is needed.
  4. Further, a definition of (keyword) “place attachment” is needed: A person feeling of place attachment when he/she lives in a place since years. In this case authors speak about competition among cities so that feeling is true only for young people that no move in another city for a better life.
  5. What is RMB (line 268)?
  6. What is AMOS 24.0 (line 340)?
  7. What is X2/df= (line 349)?
  8. What is RMSEA (line 349)?
  9. What is IFI= (line 349)?
  10. What is CFI=0.918) (line 349)?
  11. It is very hard to follow the text and the sequences of Tables and Models (1-10). Author have to check and try to clearly show results better referring to Tables and Models in the text.

Round 2

Reviewer 3 Report

I thank Authors for their responses.

The difference between “urban settlement intentions” and "place attachment" is still explicitly not clear. Too many statistics indicators do to lost the red line of the discourse.

Author Response

Dear Reviewer:

Thank you for your comments again. The main corrections and responses to your comments are as flowing:

The difference between “urban settlement intentions” and "place attachment" is still explicitly not clear.

Response: Scholar Weng and McElroy(2010) proposed the concept of talents’ urban settlement intentions, that is, the talents’ willingness to work and live in a city for a long time. From this concept, we can regard “settlement intentions” as a kind of attitude and behavior tendency, it reflects the comprehensive evaluation of talents in various aspects of a city and is a good indicator of the city’s ability to retain talents.

Place attachment is the emotional bond and psychological identity between an individual and a specific environment (i.e., place). It is a positive emotional tie. Place attachment may arise from cognition (i.e., the more you know about a place, the more you love it), well-known or reciprocal social networks, or a special emotional connection (i.e., a person may have “place attachment” to a place because of a fairy tale related to it.

According to your comment, we added these interpretations in corresponding paragraphs marked with blue color. Thank you very much for constructive suggestion.

All the best,

Yours sincerely,

Xuewen Li